# Bacterial behavior in human blood reveals complement evaders with some persister-like features

**Stéphane Pont**[1], **Nathan Fraikin**[2], **Yvan Caspar**[3,4], **Laurence Van Melderen**[2], **Ina Attrée**[1]*, **François Cretin**[1]*

1 Université Grenoble Alpes, Bacterial Pathogenesis and Cellular Responses team, CNRS ERL5261, CEA IRIG-BCI, INSERM UMR1036, Grenoble, France, 2 Université Libre de Bruxelles, Department of Molecular Biology, Cellular & Molecular Microbiology, Gosselies, Belgium, 3 Centre Hospitalier Universitaire Grenoble Alpes, Laboratoire de bactériologie-hygiène hospitalière, Grenoble, France, 4 Université Grenoble Alpes, CNRS, Grenoble INP, TIMC-IMAG, Grenoble, France

* francois.cretin@cea.fr (FC); ina.attree-delic@cea.fr (IA)

**Data Availability Statement:** All data are in the manuscript and its supporting information files.

**Funding:** This work was supported by grants from the French National Agency for Research (ANR-15-

## Abstract

Bacterial bloodstream infections (BSI) are a major health concern and can cause up to 40% mortality. *Pseudomonas aeruginosa* BSI is often of nosocomial origin and is associated with a particularly poor prognosis. The mechanism of bacterial persistence in blood is still largely unknown. Here, we analyzed the behavior of a cohort of clinical and laboratory *Pseudomonas aeruginosa* strains in human blood. In this specific environment, complement was the main defensive mechanism, acting either by direct bacterial lysis or by opsonophagocytosis, which required recognition by immune cells. We found highly variable survival rates for different strains in blood, whatever their origin, serotype, or the nature of their secreted toxins (ExoS, ExoU or ExlA) and despite their detection by immune cells. We identified and characterized a complement-tolerant subpopulation of bacterial cells that we named "evaders". Evaders shared some features with bacterial persisters, which tolerate antibiotic treatment. Notably, in bi-phasic killing curves, the evaders represented 0.1–0.001% of the initial bacterial load and displayed transient tolerance. However, the evaders are not dormant and require active metabolism to persist in blood. We detected the evaders for five other major human pathogens: *Acinetobacter baumannii*, *Burkholderia multivorans*, enteroaggregative *Escherichia coli*, *Klebsiella pneumoniae*, and *Yersinia enterocolitica*. Thus, the evaders could allow the pathogen to persist within the bloodstream, and may be the cause of fatal bacteremia or dissemination, in particular in the absence of effective antibiotic treatments.

## Author summary

Blood infections by antibiotic resistant bacteria, notably *Pseudomonas aeruginosa*, are major concerns in hospital settings. The complex interplay between *P. aeruginosa* and the innate immune system in the context of human blood is still poorly understood. By studying the behavior of various *P. aeruginosa* strains in human whole blood and plasma, we

CE11-0018-01), the Laboratory of Excellence GRAL, funded through the University Grenoble Alpes graduate school (Ecoles Universitaires de Recherche) CBH-EUR-GS (ANR-17-EURE-0003), and the Fondation pour la Recherche Médicale (Team FRM 2017, DEQ20170336705) to I.A. S.P. was awarded a Ph.D. fellowship from the French Ministry of Education and Research, and acknowledges the European Molecular Biology Organization for the short-term fellowship enabling 2-month stay in L. van Melderen's laboratory allowing further characterization of the evader population. Institutional funding support from CNRS, INSERM, CEA, and Grenoble Alpes University is also acknowledged. The funders had no role in study design, data collection or analysis, decision to publish, or preparation of the manuscript.

**Competing interests:** The authors have declared that no competing interests exist.

showed that bacterial strains display different rate of tolerance to the complement system. Despite the complement microbicide activity, most bacteria withstand elimination through phenotypic heterogeneity creating a tiny (<0.1%) subpopulation of transiently tolerant evaders able to persist in plasma. This phenotypic heterogeneity thus prevents total elimination of the pathogen from the circulation, and represents a new strategy to disseminate within the organism.

## Introduction

The incidence of bacterial bloodstream infections (BSI) in high-income countries is as extensive as that of strokes, ranging from 113 to 204 cases per 100,000 inhabitants [1]. BSI, whether nosocomial or community-acquired, have poor prognosis, with mortality rates up to 40% [1,2]. They are also a leading cause of healthcare-associated infections in intensive care units (ICUs) [3] and neonatal wards [4], and are particularly prevalent in elderly patients (6,000 cases per 100,000 population). In this demographic, mortality rates can be up to 70% [5]. Their extensive impact on overall hospital costs (>$40,000 per patient in the US) [6] make BSI a major public health concern.

Many different bacterial species can cause BSI, among which *Escherichia coli*, *Staphylococcus aureus*, *Klebsiella* species, *Pseudomonas aeruginosa*, *enterococci*, *streptococci* and coagulase-negative *staphylococci* are the most prominent [1]. *P. aeruginosa* is mainly associated with nosocomial infections, and bacteremia caused by this pathogen has a poor prognosis [2,7,8] with very high mortality rates [9]. *P. aeruginosa* can survive in many different environments and colonizes plants, animals, and humans [10]. In addition to BSI, it is responsible for a number of life-threatening complications including acute pneumonia and skin infection in immuno-compromised and elderly patients, as well as degradation of lung function in chronically-infected cystic fibrosis patients [11]. The major health concerns related to *P. aeruginosa* are linked to intrinsic and acquired resistance to currently available antibiotics [12].

The capacity of *P. aeruginosa* to survive in the human body hinges on a balance between its numerous virulence factors and the presence of multiple host-defense mechanisms. Regardless of the primary site of infection, *P. aeruginosa* can cross the epithelial and endothelial barriers to reach the bloodstream [13,14]. In the blood, the bacteria encounter the innate immune system, composed essentially of neutrophils, monocytes, and the complement system. Interactions between *P. aeruginosa* and this innate immune system have mainly been studied using selected strains and purified components, such as isolated complement proteins or phagocytes, or serum [15–20], in conditions differing from those found in the human blood [21,22]. Recent data indicated that systemic *P. aeruginosa* infection could lead to pathogen transmission as the bacteria were found to disseminate and propagate through the gallbladder and intestinal tract in a murine model of infection [23]. However, the mechanisms allowing bacteria to persist in the blood remained unclear. Within this study, we examined the behavior of a number of laboratory and recently isolated clinical *P. aeruginosa* strains in a standardized assay using fresh whole blood from healthy donors. Our results showed that, although complement exerts an essential antibacterial activity in the blood, individual bacterial strains display variable levels of tolerance. We evidenced, even for the most sensitive strains, the characteristic biphasic killing curves reminiscent of antibiotic persisters, and characterized a small subpopulation of phenotypic variants that we named complement evaders. These rare cells withstand complement-mediated lysis through phenotypic heterogeneity. Moreover, we discovered that several other major Gram-negative human pathogens shared the same capacity to escape

human complement by forming intrinsically plasma-resistant evaders. Complement evaders may have a very significant impact on bacterial dissemination.

## Results

### Laboratory and clinical strains display diverse survival rates in human blood

Using a highly standardized method in human whole blood (HWB), we examined the survival of six *P. aeruginosa* strains with distinct toxin repertoires and serotypes (Fig 1). These assays included the commonly used laboratory strains PAO1 [24], PA14 [25] and PA7 [26], which belong to distinct phylogenetic lineages/groups. PAO1 and PA14 both possess the type III secretion system (T3SS) which they use to translocate ExoS or ExoU, respectively, into target host cells. PA7 lacks T3SS genes, but encodes the pore-forming toxin Exolysin A [26,27]. In addition to these laboratory strains, we included in the survey three *P. aeruginosa* strains recently isolated from infected patients [27–29]. *E. coli* CF7968, a derivative of the K12 laboratory strain [30] was added as a control (S1 Table). Bacteria were incubated for 3 h in HWB from healthy donors, and bacterial survival was assessed in ten independent experiments by counting colony-forming units (CFU) (Fig 1A). Strains showed clearly distinguishable and reproducible survival rates. Approximately 10% of the laboratory and reference strains, PAO1 (ExoS$^+$) and PA7 (ExlA$^+$), survived, whereas only around 1% of the most sensitive strain, PA14 (ExoU$^+$), was still present following the 3-h incubation in HWB. Very different survival rates were measured for the three recent clinical strains, YIK (ExoU$^+$), CLJ1 and IHMA87 (ExlA$^+$), ranging from 0.05% for CLJ1 to complete tolerance for YIK. Only the non-pathogenic laboratory strain *E. coli* CF7968 was completely eliminated, with no detectable CFU after 3 h exposure to HWB. The nature of the strain's virulence factors (T3SS versus ExlA) did not

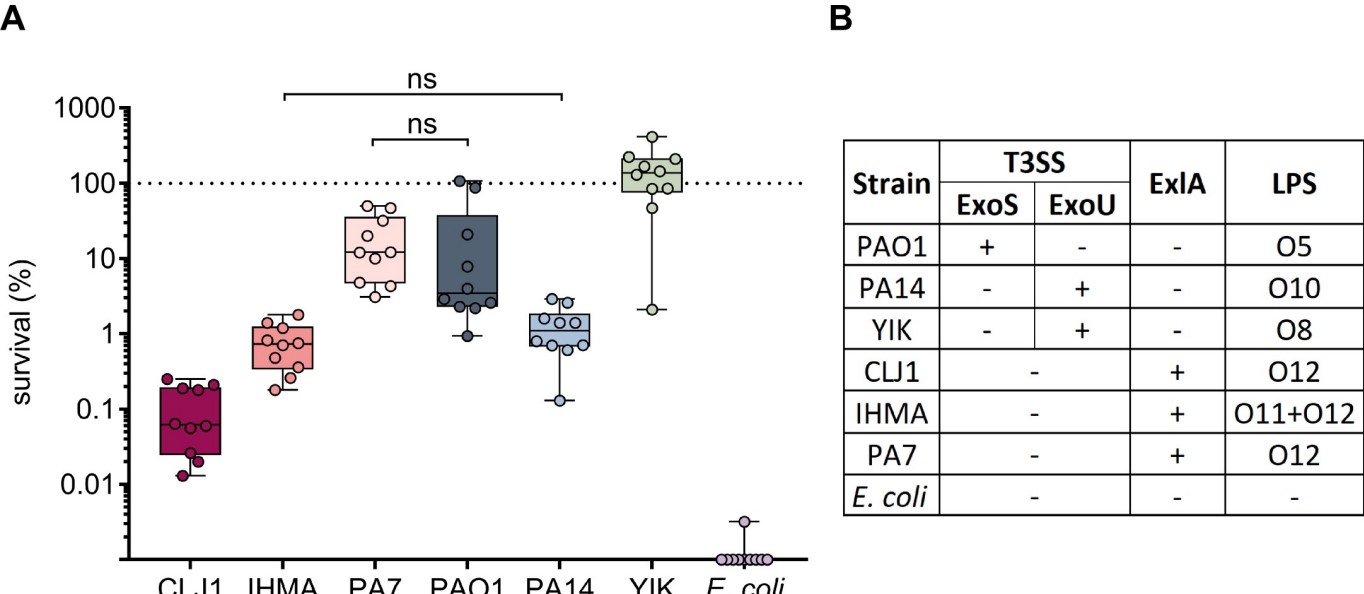

**A**

**B**

| Strain | T3SS | | ExlA | LPS |
|--------|------|------|------|-----|
| | ExoS | ExoU | | |
| PAO1 | + | - | - | O5 |
| PA14 | - | + | - | O10 |
| YIK | - | + | - | O8 |
| CLJ1 | - | | + | O12 |
| IHMA | - | | + | O11+O12 |
| PA7 | - | | + | O12 |
| *E. coli* | - | | - | - |

**Fig 1. *P. aeruginosa* strains display various rates of resilience in the HWB model. (A)** *P. aeruginosa* survival in blood is independent of strain origin or the secreted toxin profile (ExoU, ExoS, or ExlA). Exponentially-growing bacteria were incubated for 3 h in blood obtained from 10 different healthy donors (n = 10), and bacterial survival was determined by serial dilutions and CFU counting. ns: non-significant. When not stated, differences in survival were significant. Kruskal-Wallis test, p <0.001; Student-Newman-Keuls post-hoc test: p <0.05. Dots reported on the x-axis correspond to no detectable colonies. **(B)** Nature of the toxins secreted by each strain, and serotypes of these bacteria. Note that the IHMA87 strain cross-reacts with the two antisera O11 and O12, while *E. coli* CF7968 lacks O antigens.

appear to confer any significant benefit for survival, as similar sensitivities were measured for ExlA[+] and T3SS[+] strains (e.g. IHMA87 versus PA14, or PA7 versus PAO1). In addition, survival in HWB did not correlate with a given serotype, as highly variable survival rates were measured for the three O12 strains (Fig 1B). Lack of the O antigen was detrimental for the bacteria, as illustrated by the hypersensitivity of *E. coli* CF7968.

### Intrinsic *P. aeruginosa* tolerance in blood is linked to complement activity, through either MAC insertion or opsonophagocytosis

To explore the origin of the extensive differences in survival measured in HWB, we first determined how well each strain was recognized by immune cells. While YIK induced marginal TNFα and IL6 production, all the other strains tested triggered similar high levels of cytokines (S1 Fig). The toxins ExoS, ExoU, and Exolysin A are known to induce apoptosis or necrosis in a variety of eukaryotic cells, including white blood cells which may play a role in bacterial clearance from the blood [16,31–34]. We therefore examined the cytotoxic potential of each strain toward purified circulating leukocytes by monitoring lactate dehydrogenase (LDH) release. YIK induced a complete elimination of neutrophils, whereas most other strains showed similar levels of cytotoxic potential with no significant differences (S2A Fig). Every tested strain appears more efficient at killing neutrophils than mononucleated cells (S2B Fig). Thus, except for YIK, the extent to which the bacterial strains tested here were recognized by and destroyed circulating leukocytes could not explain the different survival rates measured in HWB.

We next assessed the capacity of the strains to cope with bactericidal effectors present in plasma (Fig 2A). As with bacterial survival in HWB, in plasma the survival rates for the six selected *P. aeruginosa* strains revealed a range of sensitivities from resistance to almost complete eradication. In agreement with previous reports [17,35,36], the two laboratory strains PAO1 and PA14 had contrasting survival patterns: PAO1 was tolerant, whereas PA14 was sensitive. The strains that were most tolerant to HWB (PA7, PAO1 and YIK) were fully resistant to plasma-mediated killing. The similar survival profiles between HWB and plasma, and the fact that survival of the plasma-sensitive strains CLJ1, IHMA87, and PA14 was 10-fold lower in plasma than in whole blood suggest an important role for humoral immune effectors in bacterial clearance within HWB. Due to its complete resistance to killing, the YIK strain was excluded from subsequent experiments.

To assess how fluid-phase effectors contribute to bacterial elimination in HWB, we examined the survival of the different strains by counting CFUs following incubation in HWB reconstituted after heat-inactivation of the plasma (see Methods). Heat-inactivation, which eradicates complement activity, resulted in full survival of all strains, including the sensitive *E. coli* CF7968 (Fig 2B). Indeed, most strains multiplied when incubated in blood with heat-inactivated plasma, as observed by the increased CFUs compared to the starting population.

As heat-inactivation of plasma may inhibit different bactericidal plasmatic components [37], and to address whether the elimination of sensitive strains in HWB specifically relies on the complement system or other circulating effectors with antimicrobial properties, we used specific complement inhibitor OmCI (also known as Coversine) [38]. OmCI inhibits the cleavage of C5, hampering membrane attack complex (MAC) formation and subsequent complement-mediated lysis [37]. Addition of OmCI in plasma resulted in complete rescue of *P. aeruginosa* strains CLJ1, IHMA87 and PA14 from killing, demonstrating that the *P. aeruginosa* elimination in HWB is solely dependent on complement microbicide activity (Fig 2C).

The antimicrobial activity of complement relies on three main mechanisms: i/ bacterial lysis due to the insertion of the MAC into the bacterial envelope [39], ii/ opsonophagocytosis,

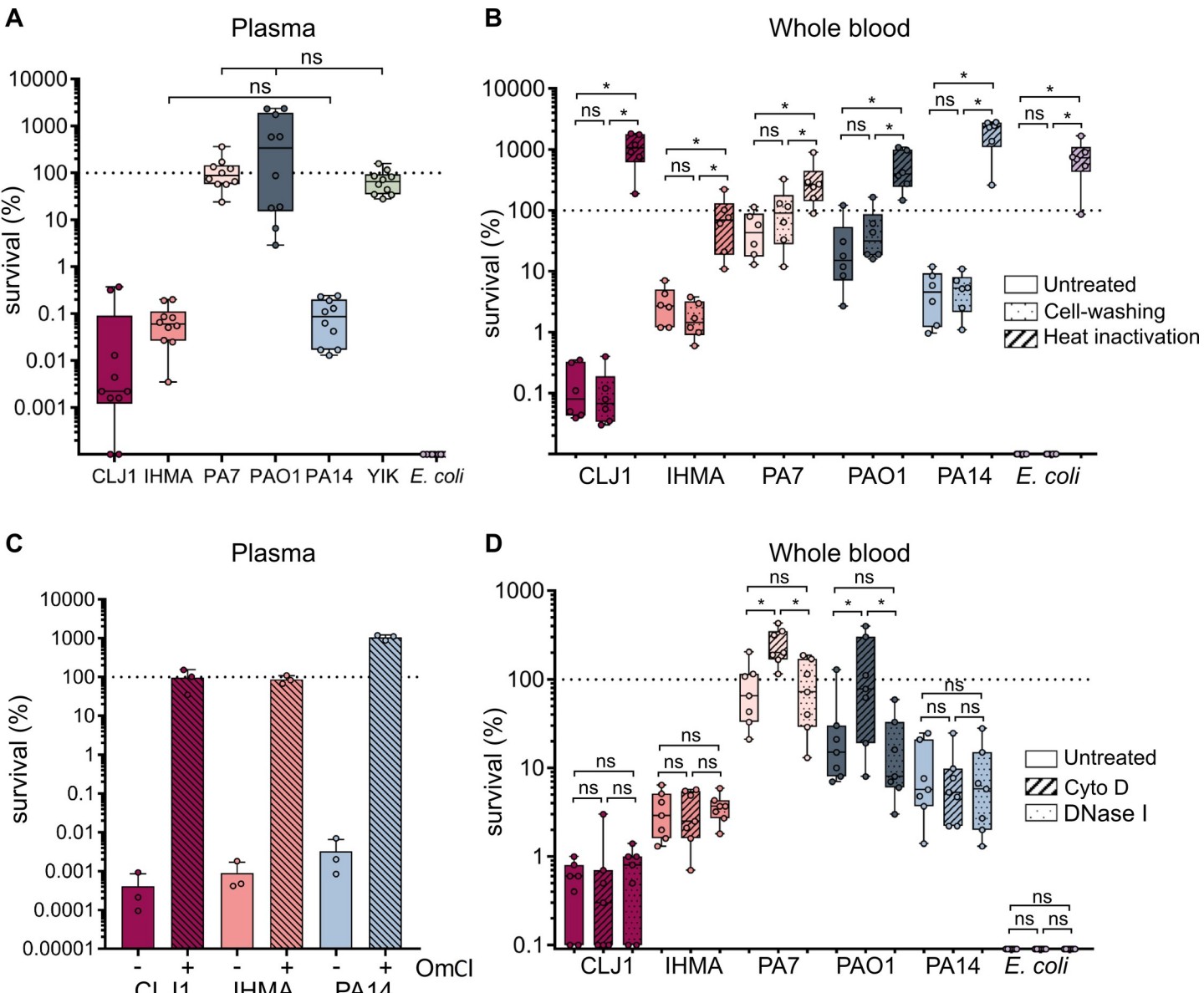

**Fig 2. Complement is the main immune factor involved in bacterial clearance in HWB. (A)** Survival of *P. aeruginosa* strains in human plasma. Strains were incubated in plasma from different healthy donors (n = 10), and survival was measured based on CFU counts. ns: non-significant. When not stated, differences in survival were significant. Kruskal-Wallis test, p <0.001; Student-Newman-Keuls post-hoc test: p <0.05. Note similarities in survival profiles to those shown in Fig 1. **(B)** Heat-treating plasma prevents elimination of bacteria from HWB. Bacteria were incubated for 3 h in HWB from different donors (n = 6) with (hashed) and without (solid) heat-treatment to inactivate complement, or only with washed blood cells without plasma heating (dotted). The effect of this treatment on bacterial survival was assessed based on CFU counts. **(C)** Inhibition of MAC formation rescues *P. aeruginosa* strains from killing. Bacteria were incubated in a pool of human plasma for 3 h with (hashed) or without (solid) OmCI at 20 μg/mL (IHMA87 and PA14) or 40 μg/mL (CLJ1) and survival was measured based on CFU counts. Due to bacterial clumping, sonication was used to increase the accuracy of CFU measurement of IHMA87, as described in Methods. Data represent mean ± SD of three independent experiments. **(D)** Phagocytes are involved in the elimination of a limited number of strains. Bacteria were incubated for 3 h in HWB from different donors (n = 7) in the absence (solid) or presence of Cytochalasin D (hashed) or DNase I (dotted), to monitor the impact of these treatments on strain survival. **(B)** and **(C)**: Kruskal-Wallis test, p <0.05; Student-Newman-Keuls post-hoc test: * p <0.05. Dots on the x-axis correspond to no detectable colonies.

which combines C3b binding at the pathogen's surface and its recognition by complement receptors [39], and iii/ formation of neutrophil extracellular traps (NETs) to trap and kill invading bacteria [40]. To elucidate which of these mechanisms was involved in bacterial clearance, we treated the blood with Cytochalasin D to inactivate phagocytosis, or with DNase I to

prevent NETs formation [22,41] and monitored bacterial survival. Following treatment with DNase I, the same level of bacterial elimination was observed as with untreated HWB (Fig 2D), indicating that NETs play a negligible role in the process observed here. In contrast, PA7 and PAO1 elimination appear to involve some internalization by phagocytes, as their survival was consistently increased in Cytochalasin D-treated blood (Fig 2D). Because the clearance of these two strains was also complement-dependent, we conclude that they are eliminated through opsonophagocytosis. Blocking phagocytosis had no impact on the other complement-sensitive strains CLJ1, IHMA87, PA14, and *E. coli* CF7968, suggesting that they are killed through direct MAC-induced lysis.

Importantly, even though most of the strains were highly sensitive to complement, we recurrently detected a subpopulation of survivors corresponding to <1% or even 0.002% of the initial inoculum, in HWB and plasma, respectively (Figs 1A and 2A). These results suggest that a small bacterial subpopulation, that we termed "evaders", differ in phenotype from the majority of the population, and display increased tolerance to complement-mediated killing. The level of residual surviving cells appears higher (by 1-log) in whole blood than in plasma (Figs 1A and 2A). This higher clearance of bacteria in plasma than in HWB was already observed in the case of *Salmonella typhimurium* [42]. A possible explanation could be that within HWB, complement activity is more strongly repressed due to all membrane-bound negative regulators at the surface of circulating leukocytes (i.e CD59, CR1, DAF), while absent in plasma.

## Complement evaders display persister-like features

As indicated above, only the laboratory strain *E. coli* CF7968 was entirely eliminated upon exposure to plasma. The subpopulation of *P. aeruginosa* evaders in this environment ranged from 0.1% down to 0.002% of the initial bacterial load, depending on the strain (Fig 2A). We further investigated this intriguing difference in sensitivities using the three complement-sensitive strains PA14, CLJ1, and IHMA87, by carefully examining the kinetics of bactericidal activity in plasma over a 6-h incubation (Fig 3A). A biphasic curve of bacterial killing was observed, with the majority of the sensitive population (> 99.9%) eliminated within 2 h. Following this first phase, killing slowed down, reached a plateau and left a minor subpopulation of surviving cells. However, this subpopulation failed to grow, even after 6 h. We verified that the drop off in killing rate was not due to depletion of complement activity after 2 h by retesting the used plasma. The used plasma was still sufficiently active to kill $> 10^7$ *P. aeruginosa* PA14 cells during a 1-h incubation (S3A Fig). Thus, the plasma had a residual bactericidal capacity, sufficient to eliminate a population at least 4-log more numerous than the number of evaders. As further evidence that evaders are not simply a result of bacterial overload of the complement system, inoculating plasma with 10-fold fewer bacterial cells resulted in the same proportion of evaders (S3B Fig). Based on these results, evaders correspond to phenotypic variants displaying complement tolerance, and appear to be present in similar proportions to antibiotic-tolerant persisters [43,44].

To further phenotypically characterize evaders, we re-cultured the survivors recovered from a first incubation in plasma and challenged their progeny. As shown in Fig 3B, following re-culture, the bacterial population had a similar sensitivity profile to previously unchallenged cells. In some cases, the number of evaders in these repeat challenges was below the limit of detection in our experimental settings (e.g. CLJ1 after 2 h). This apparent concern is a hallmark in antibiotic persisters research [45]. Thus, neither evaders nor persisters reflect the emergence of resistant mutants, rather the evaders' phenotype, similar to persisters', is transient and reversible.

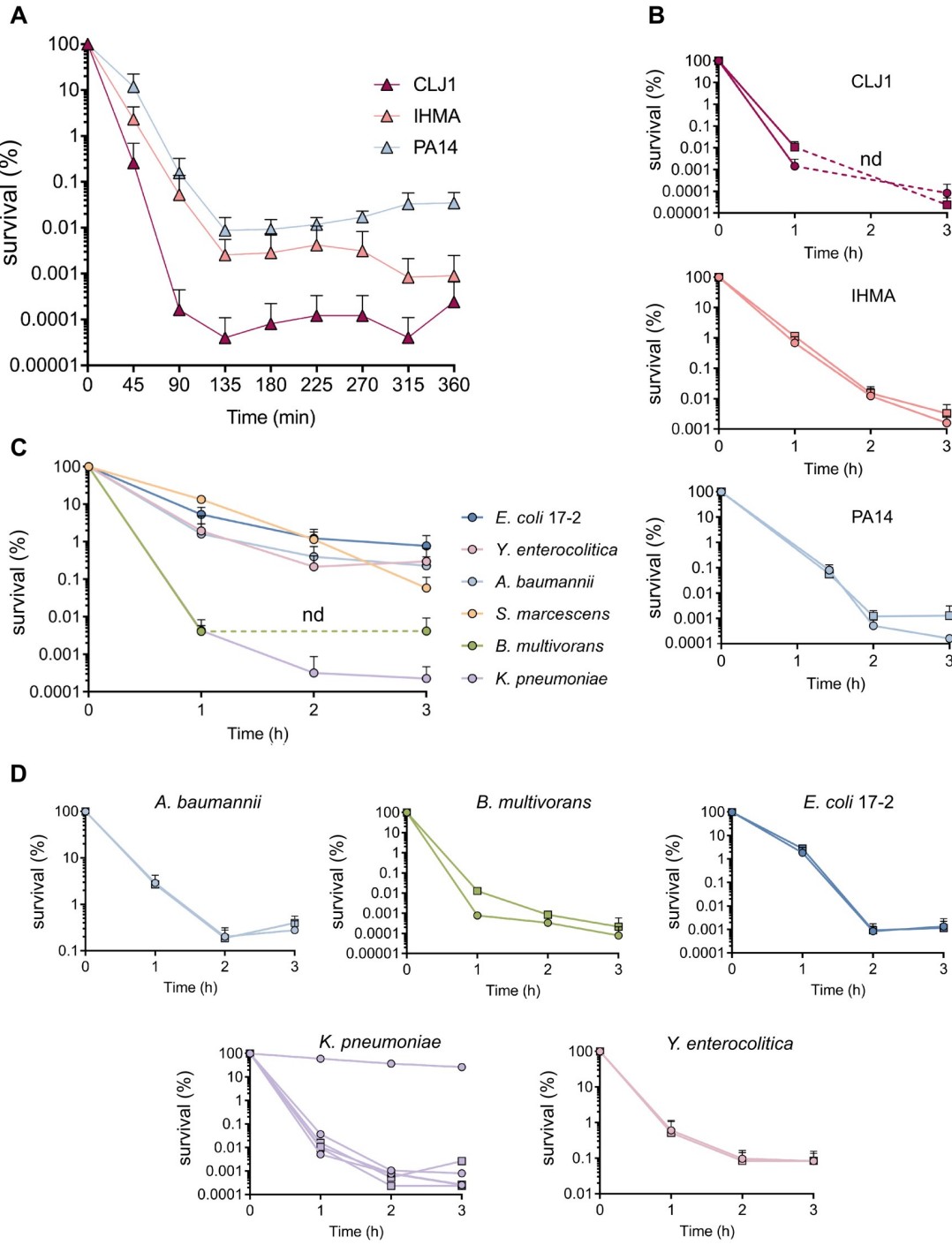

**Fig 3. Evaders are rare, complement-resistant, phenotypic variants.** (**A**) Prolonged exposure of *P. aeruginosa* to plasma results in killing curves reminiscent of those that characterize persister formation. The three complement-sensitive strains PA14, CLJ1 and IHMA87 were incubated for 6 h in a pool of human plasma, and their survival was measured every 45 min to determine the kinetics of their survival. (**B**) The evader phenotype is reversible and not the result of fixed mutations. Following a first challenge, a single "evader" colony was re-cultured to assess the survival of its progeny (circle) in pooled plasma during 3 h. Survival was compared to that of a population that was never exposed to complement (square). (**C**) Evaders are a common feature among Gram-negative bacteria. *A. baumannii*, *B. multivorans*, enteroaggregative *E. coli*, *K. pneumoniae*, and *Y. enterocolitica* were incubated in pooled human plasma for 3 h, and their survival was assessed hourly. (**D**) The reversibility of the tolerant phenotype was assessed in individual colonies as in (**B**). Data represent mean ± SD of three independent experiments (**A** to **D**). Note that for *K. pneumoniae* in (**D**), the three experimental points were not pooled as one resistant mutant was isolated, presenting a survival profile different from the two other evader clones. nd: non-detected.

As complement evaders were detected for all sensitive *P. aeruginosa* strains, we tested whether this behavior could be extended to other Gram-negative bacteria. To that aim, we selected strains of seven Gram-negative species (S1 Table) and assessed their survival in pooled plasma. Among the strains tested, *Stenotrophomonas maltophilia* was undetectable after 1 h of incubation, and *Serratia marcescens* presented what we called a tolerant phenotype in plasma, with a slow but constant elimination rate. In contrast, the five other strains tested–*Acinetobacter baumannii*, *Burkholderia multivorans*, enteroaggregative (EA) *E. coli* 17–2, *Klebsiella pneumoniae*, *and Yersinia enterocolitica*–presented a biphasic survival curve similar to the one recorded for *P. aeruginosa* (Fig 3C). For these strains, the proportion of evaders withstanding plasma-mediated killing after 3 h of incubation ranged from 1% to 0.0002% of the initial population. When individual evader colonies were re-cultured, a population as sensitive as the parental one was recovered, as seen for *P. aeruginosa*. Unexpectedly, for *K. pneumoniae*, one evader colony out of the three that were randomly selected gave rise to a resistant population (Fig 3D). Thus, plasma-resistant mutants can be selected and we do not currently know whether selection occurs over the course of the pre-challenge culture steps, or during contact with plasma.

As with *P. aeruginosa*, we verified that complement was the main driver of the elimination of these plasma-sensitive strains by using OmCI. Prevention of MAC formation abolished elimination of *A. baumannii*, *E. coli* 17–2 and *Y. enterocolitica*, whereas it only partially rescued *B. multivorans* and *K. pneumoniae*, for which the number of surviving cells increased by 190 and 40 times, respectively. However, their overall survival was still below 0.05% (S4 Fig). We conclude that depending of the strain, plasma evaders are either exclusively or only partly composed of complement-tolerant bacteria, the later observation suggesting that other fluid-phase effectors can be involved in the selection of this subpopulation.

## Evaders are not dormant cells

To further describe the features of complement evaders, we assessed if these rare bacteria could arise in conditions known to trigger the emergence of antibiotic-tolerant persisters. Antibiotic persisters can correspond to up to 100% of cells in the stationary phase of growth [46], presumably due to growth arrest [47–49]. Thus, after verifying bacterial growth-rates and states (log versus stationary) (S5 Fig), we tested whether the evaders observed in exponentially growing cultures corresponded to residual non-growing cells from the previous overnight culture, or to rare cells that had already entered the stationary state after a few hours of culture. To eliminate possible artefacts due to the growth phases [45,50], we challenged stationary-phase cells or exponentially growing cells with plasma. For the three strains *P. aeruginosa* IHMA87, *A. baumannii* and *Y. enterocolitica*, we observed that the level of evaders was higher in actively growing cultures, while for the other strains their proportions were similar between the two states (Fig 4A). Therefore, exponentially growing bacteria had a similar or higher capacity to produce evaders, suggesting that the emergence of evaders is unrelated to dormancy before complement challenge. To assess whether metabolic shut-down could increase the proportion of evaders, prior to exposure to plasma, *P. aeruginosa* IHMA87 was treated with the protonophore cyanide m-chlorophenylhydrazone (CCCP), which uncouples oxidative phosphorylation. Exposure to CCCP has been reported to increase the proportion of antibiotic persisters in *P. aeruginosa* by almost 2-log [51]. The addition of CCCP had no effect on bacterial survival in LB (S6 Fig), but its use before the plasma challenge completely abolished the detection of evaders after 2 h of incubation (Fig 4B), suggesting that proton-motive force is necessary for survival. To assess more growth parameters of evaders and the possibility that they display a "slow growth phenotype", we performed time-lapse microscopy of IHMA87-GFP bacteria on agarose pads after incubation in normal or heat-inactivated plasma

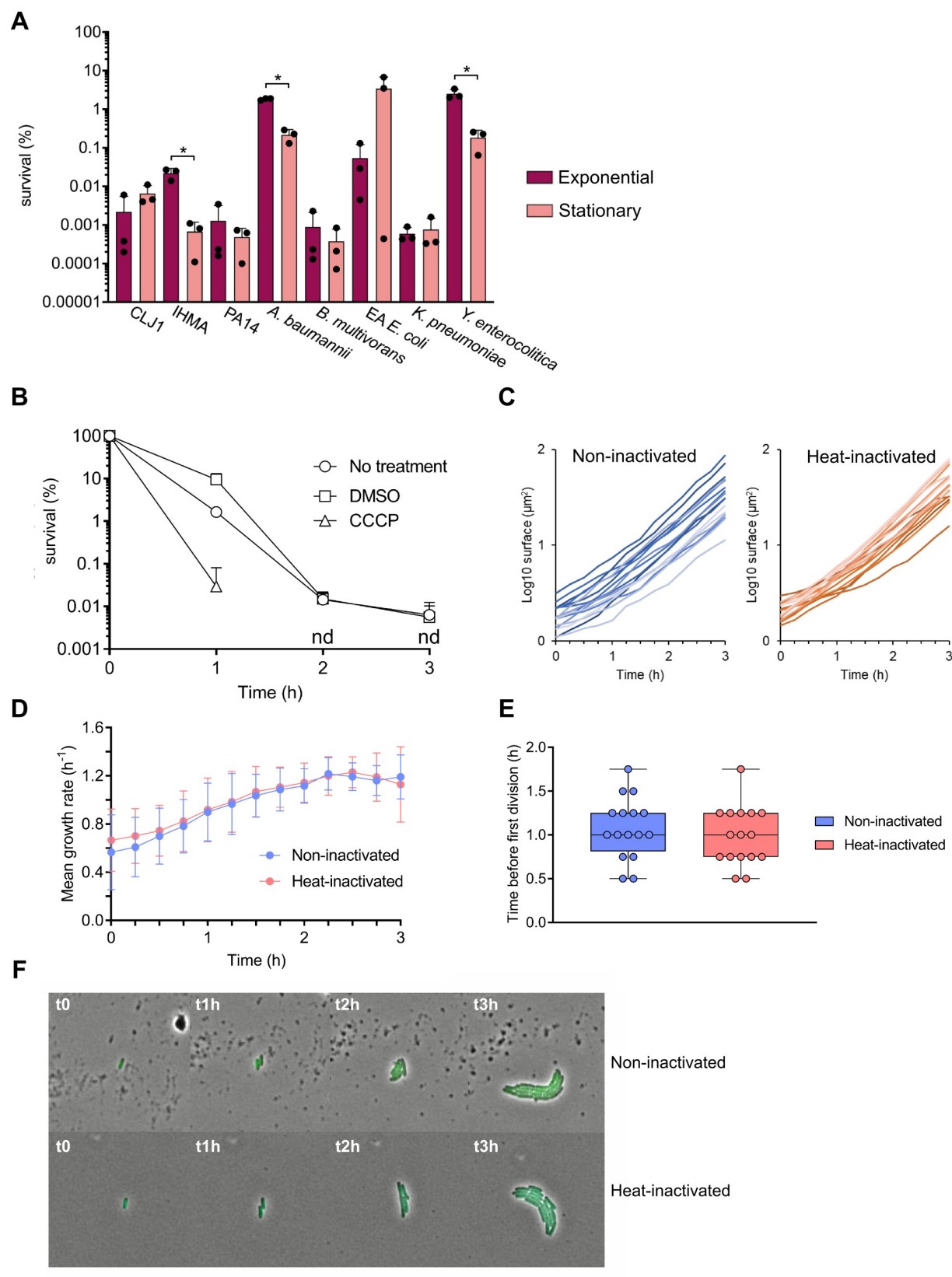

**Fig 4. Complement evaders differ from antibiotic persisters.** (**A**) Cells in the stationary phase do not form more evaders than exponentially growing bacteria. Bacteria from stationary phase and exponentially growing cultures from every evader-forming strain were challenged in a pool of plasma for 3 h to compare their ability to form evaders. Student's t-test: * p <0.01, performed on log-transformed values. (**B**) Formation of evaders requires active metabolism. Survival kinetics of IHMA87 in a pool of plasma, following 1-h treatment with the protonophore CCCP in LB. As controls, bacteria in LB were either not treated or incubated 1 h with 0.5% DMSO prior the challenge. nd: non-detected. (**C**) **to** (**E**) Following stress removal, evaders show no growth defect. After 3 h incubation of IHMA87-GFP in pooled plasma following heat-inactivation or not, surviving cells were recovered and spotted on a 2% agarose pad containing LB, allowing the bacteria to regrow. Using time-lapse microscopy, individual microcolony surface (**C**), mean growth rate (**D**) and elapsed time before the first division (**E**) were determined. From two independent experiments, 16 evader cells were analyzed and compared with 16 non-evader cells from the control condition (heat-inactivated plasma). (**F**) Time-lapse microscopy of two representative microcolonies from each condition. Data represent mean ± SD of three independent experiments (**A** and **B**).

(Fig 4C–4F). Our results show that evaders immediately and actively grow following plasma removal and spotting on agarose pads, forming exponentially-growing microcolonies (Fig 4C). In both normal and heat-inactivated plasma-treated conditions, growth rate was low upon spotting ($0.6\pm0.3$ h$^{-1}$) but stabilized at levels similar to batch cultures ($1.2\pm0.1$ h$^{-1}$) after 2.5 h (Fig 4D). Division was not impaired in evaders as the mean time to reach a first division did not vary between normal ($1.08\pm0.35$ h) and heat-inactivated plasma-treated cells ($0.98\pm0.34$ h) (Fig 4E and 4F). Altogether, these results suggest that survival to plasma treatment is not dependent on growth rate but requires energy and active metabolism.

## BSI isolates show full range of plasma resistance and presence of evaders

As we had demonstrated that the central driver of bacterial clearance from blood was the complement system, mainly through its direct lytic activity, we next investigated survival in plasma of a cohort of twelve clinical strains isolated from patients with BSI (S1 and S2 Tables) to determine their capacity to form evaders. Like the data obtained with the initial six selected strains, BSI isolates displayed differences in survival rates in plasma, of up to five orders of magnitude. Four strains (PaG2, PaG5, PaG6, and PaG10) were tolerant to killing, displaying > 50% survival, with some even able to multiply in these conditions (PaG2, PaG6 and PaG10) (Fig 5A). In contrast, for other strains (PaG8, PaG9, PaG14) just < 0.02% of the initial population survived (Fig 5A). The limited number of strains and the high diversity of serotypes identified (S1 Table) would make any attempt to correlate bacterial survival in plasma with strain serotype too speculative. Even though some strains were highly sensitive to plasma, none was fully eliminated. To verify that these surviving cells corresponded to evaders, we also assessed their survival kinetics in plasma (Fig 5B). The three isolates PaG3, 9, and 17 presented a tolerant phenotype, with a constant rate of elimination, never reaching a plateau even after 4 h incubation. In contrast, a biphasic killing curve was recorded for the five other isolates, the kinetics of the curve varied from strain to strain, sometimes reaching a plateau after just 1 h, whereas for others the death rate started to slow from the 3-h time point. As indicated above, in some cases, surviving cells were scarce and bellow the limit of detection (e.g. PaG8 at 2 and 3 h). Thus, most clinical isolates form evaders that can withstand the bactericidal property of plasma, suggesting that this phenomenon could be exploited by *P. aeruginosa* to persist within the bloodstream in clinical settings.

## Discussion

Bacteremia caused by the multi-drug resistant opportunistic pathogen *P. aeruginosa* presents a particular threat to hospitalized patients. To gain more knowledge on host and pathogen strategies associated with BSI, we undertook an extensive analysis of a cohort of *P. aeruginosa* strains in an *ex vivo* HWB model of infection. Although many previous studies have addressed bacterial transmigration across epithelial and endothelial barriers, the interplay between the immune system and bacterial survival in HWB has been less extensively documented. We

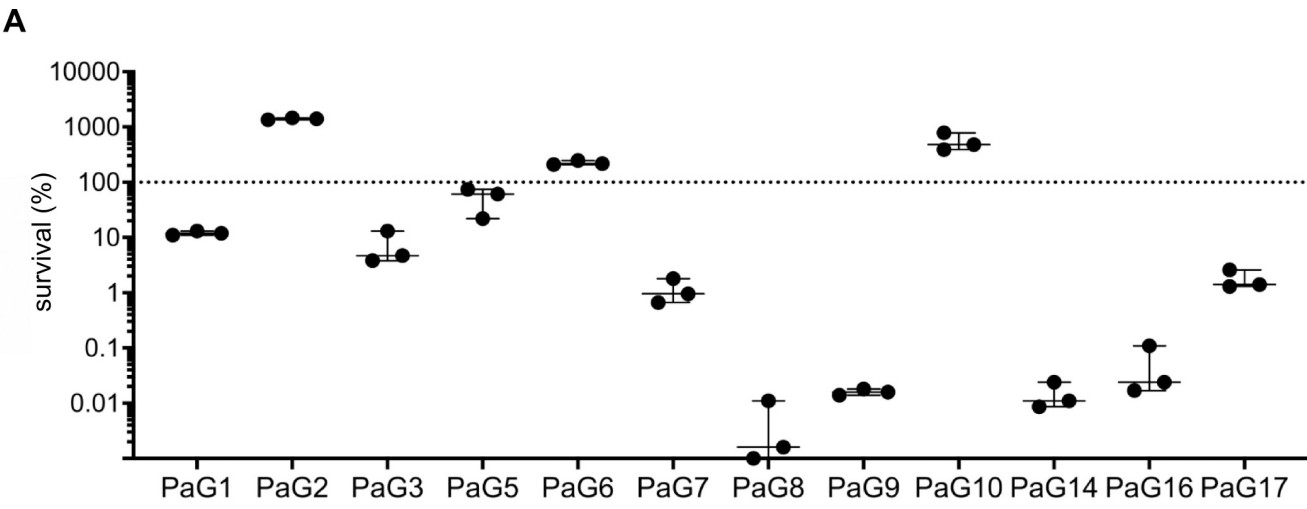

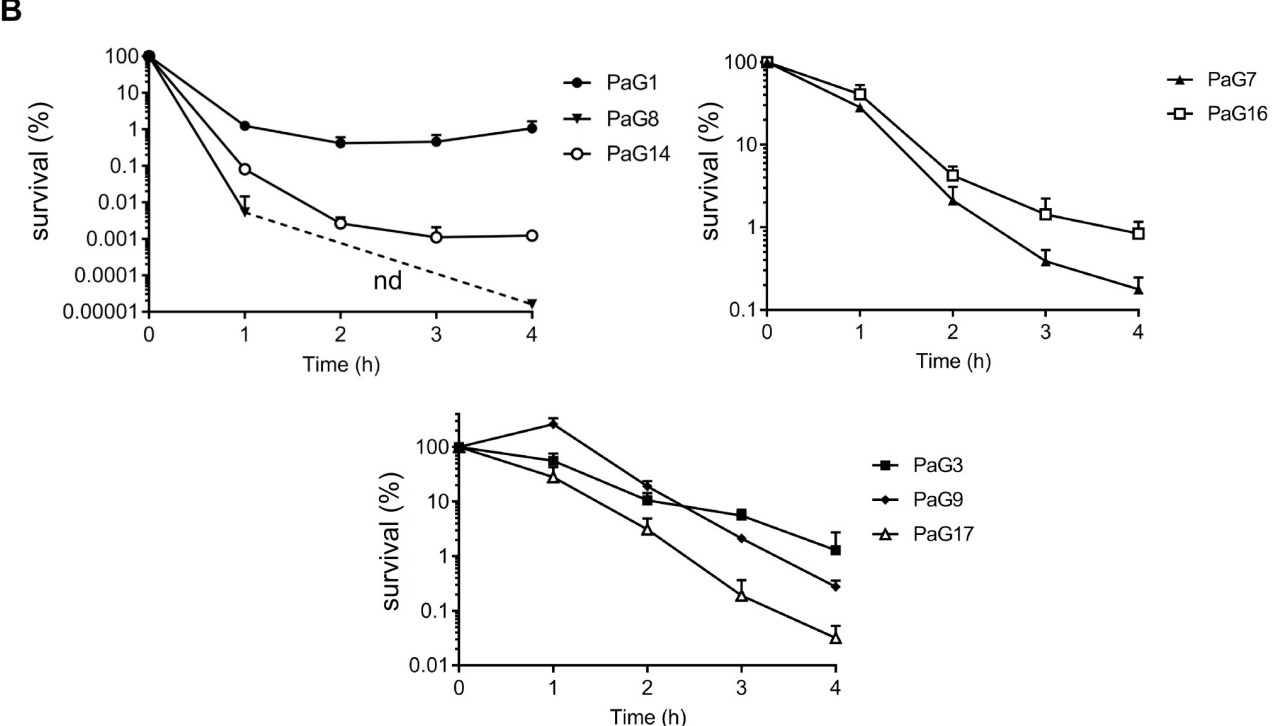

**Fig 5. BSI isolates avoid elimination mainly through complement evaders. (A)** Complement-resistance is not common to all BSI isolates. Twelve *P. aeruginosa* isolates from bloodstream infections were incubated in pooled plasma for 3 h, and their survival was assessed at the end of the incubation. Dots on the x-axis correspond to no detectable colonies. **(B)** Most complement-sensitive isolates form evaders. Kinetics of survival in pooled plasma for 3 h of eight complement-sensitive clinical isolates. Three different panels are used for clarity and to allow better visualization of the inflection points. nd: non-detected. Data represent mean ± SD of three independent experiments (**A** and **B**).

found that the nature of major *P. aeruginosa* toxins ExoS, ExoU or ExlA was unrelated to a strain's capacity to survive in HWB, even though these toxins play important roles in breaching epithelial and endothelial barriers [13,27,31,52,53]. The levels of bacterial survival measured in HWB were highly variable and directly related to the action of the complement system, as previously reported for carbapenem-resistant epidemic clones of *Klebsiella pneumoniae* [54]. NETs did not contribute significantly to bacterial killing. In some cases, PMNs

contributed to bacterial elimination through complement-dependent opsonophagocytosis, but this process was less efficient than direct MAC-induced lysis. In line with this observation, Thanabalasuriar and colleagues [55] observed that neutrophils had a limited capacity to phagocytose *P. aeruginosa* in the mouse lung vasculature, in contrast to PMNs recruited to the organ at the site of infection [56]. Complement resistance is the main driver of survival in the bactericidal environment that human blood represents. Nevertheless, among the tested isolates, we observed that to fully resist the immune system, the strains had to display both complement resistance and cytotoxicity toward professional phagocytes, as documented for the highly virulent YIK strain recently isolated from a 49-year-old individual with no known immunodeficiency [29]. Whether these characteristics apply to other clinical isolates able to survive in the blood should be investigated on bigger bacterial cohorts.

A plethora of previously identified bacterial factors could interfere with the complement system in the HWB model. Thus, bacteria could alter C3b binding [19,36,55,57–60], recruit negative complement regulators [18,20,61,62], break down complement proteins [15,63,64], or stabilize outer membrane integrity [65,66]. Consequently, bacterial resistance to complement activity appears to be multifactorial and strain-dependent, and should thus be investigated in a more systematic and uniform way to obtain a better overall picture of the complex interactions involved. Indeed, a very recent Tn-Seq approach performed in parallel on four *Klebsiella pneumoniae* strains revealed a few general, but mainly strain-dependent, factors contributing to complement-resistance [67].

The most prominent result from our study is the evidence that strains highly sensitive to plasma can nevertheless escape complement's bactericidal activity by forming phenotypic variants, or evaders. In some instances, complement evaders represent < 0.01% of the initial population, with survival kinetics reaching a plateau reminiscent of bacterial persisters following antibiotic challenges [45]. As with antibiotic persisters [45], we found that complement evaders lose their complement-tolerant phenotype upon elimination of the stress. However, complement evaders do not form under conditions previously described as triggering the appearance of antibiotic persisters. Indeed, persisters are more numerous in stationary-phase cultures [49], whereas the fraction of evaders is similar or even higher in exponentially growing cells. CCCP treatment, which decreases ATP production by membrane-bound ATPases as a result of inhibition of proton motive force was previously shown to increase the number of persister cells by 2-log upon challenge with ciprofloxacin [51]. Here, metabolic shut-down with CCCP prior to the plasma challenge abolished the emergence of evaders. Finally, we observed no growth defect/delay of these rare cells following removal of the stress. These results evidence that complement evaders are not in a dormant state, and that they emerge through an energy-dependent process. Based on these results, we propose that complement evaders share only some characteristics with antibiotic persisters to withstand a stress [43,68–71], however the mechanisms leading to the development of these traits remain a subject of debate [72], and might greatly depend on the nature of the challenge faced by the bacteria.

Interestingly, following serum challenge of uropathogenic *E. coli*, Putrinš and colleagues [73] reported the emergence of complement-resistant stationary-phase persisters. This group also identified a subpopulation of non-quiescent cells undergoing rapid division in serum, which could withstand complement-mediated lysis but was sensitive to antibiotics. This population may be the same as the evaders described here. However, Putrinš and colleagues only observed these cells from stationary-phase cultures, as exponentially growing bacteria appeared to be serum-resistant in their experiments. These results might suggest that, in addition to complement evaders, bacteria may have developed a number of ways to transiently hide from the immune system when present in the blood. Whether complement evaders are also antibiotic-tolerant requires further investigation.

Host immunity can amplify a pathogen's phenotypic heterogeneity, promoting the formation of antibiotic persisters both *in vivo* and *in vitro*. For example, upon lung infection with *M. tuberculosis*, cell-to-cell variations in ribosomal RNA transcription patterns increased markedly compared to growth-permissive *in vitro* conditions [74]. Bacterial uptake by macrophages was also recently shown to induce persistence in both *Salmonella enterica* and *S. aureus*, in response to the stress conditions encountered during vacuolar internalization [75–77]. Exposure to human serum has also been linked to an increased frequency of antibiotic persisters in *Vibrio vulnificus* [78]. The process involved in the emergence of this population is at least partly mediated by complement activity, as a lower proportion of surviving cells was detected following exposure to heat-inactivated serum [78]. Our results show that human plasma can trigger phenotypic diversity in addition to antibiotic persistence, although we still lack information on whether the evader phenotype emerges spontaneously (are these cells already present in the population prior to the stress?) or in response to a trigger (do they only appear upon contact with complement?) [50]. Ongoing experiments should reveal whether evaders activate less the complement system, or whether they harbor an increased tolerance to MAC insertion at their surface. As these bacterial cells also persist in blood, we hypothesize that lower amount of C3b deposits at their surface; otherwise, they should be efficiently phagocytosed by circulating neutrophils. This minor population of evaders has been ignored so far as it can represent less than 0.01% of the initial population, and because bacterial survival in serum is often monitored at a single time point (usually after a 1-h challenge), rather than examining the kinetics of survival over a long enough period to reach a plateau.

Although BSIs were historically considered "dead-ends" for infectious agents, Bachta and colleagues [23] recently reported that once in the blood, a subpopulation of *P. aeruginosa* migrates to the gallbladder in mice, where it replicates and can exit the organism through the intestinal tract, causing contamination of cage-mates. In our experiments, not all BSI isolates were complement-tolerant, but most of the sensitive strains could form evaders, making the bacteria potentially transmittable.

Numerous bacterial pathogens affecting humans may form an evader population, which could represent a reservoir of complement-tolerant cells capable of disseminating and spreading throughout the organism. By elucidating the molecular mechanisms through which complement evaders emerge, notably by performing transcriptomic/proteomic profiling of these populations, we hope to identify ways to diminish the risks of bacteremia caused by various bacterial pathogens.

## Methods

### Ethics statement

Medical data from patients infected with *P. aeruginosa* strains were extracted from medical records. No nominative or sensitive personal data were recorded, and the study only involved the reuse of already available data. This study falls within the scope of the French Reference Methodology MR-004 for studies not involving human subjects.

### Bacterial strains and culture conditions

Bacterial strains used in this study are listed in S1 Table. Bacteria were grown in liquid Lysogeny Broth (LB) prepared according to Miller's formulation (0.5% yeast extract, 1% tryptone, 10% NaCl) for >16 h with agitation (300 rpm) at 37˚C, except for *Y. enterocolitica*, which was cultured at 28˚C. If not otherwise specified, the overnight culture was diluted approx. 40 times in LB to have an initial $OD_{600nm} \sim 0.1$ and placed at 300 rpm at 37˚C until the $OD_{600nm}$ reached $\sim 1$.

## Whole blood and plasma killing assays

Heparinized HWB from healthy donors was provided by the French National blood service (EFS, Grenoble, France) and was used within 3 h of collection. Bacteria resuspended in RPMI (Thermo Fisher Scientific, Illkirch, France) were incubated in HWB (90% final blood concentration) at a theoretical multiplicity of infection (MOI) of 5 per phagocyte (monocytes and granulocytes), which corresponded to a final bacterial concentration of $2.25 \times 10^7$ mL$^{-1}$. The precise value was verified for each experiment by plating the bacteria on LB agar plates at $t_0$ and counting colony-forming units (CFU) after ~ 15 h incubation at 37˚C. Tubes were incubated for 3 h on a rotating wheel at 37˚C in a 5% $CO_2$ atmosphere. Following incubation, bacterial survival was determined following serial dilutions in $H_2O$ by colony counting on LB or selective PIA (Pseudomonas Isolation Agar) medium. Bacterial survival was expressed as a percentage (%) of survivors calculated from the CFU number after 3 h ($t_{3h}$) incubation relative to the CFU measured in the initial inoculum ($t_0$). To inhibit the potential bactericidal effect of phagocytosis and Neutrophil Extracellular Traps (NETs), Cytochalasin D (10 μM) and DNase I (200 U/mL) (Sigma-Aldrich, Saint-Quentin-Fallavier, France) were applied to the HWB for 30 min at 37˚C prior to the addition of bacteria. In the non-treated condition, RPMI medium without inhibitors was added to the blood.

To inactivate complement, HWB was centrifuged for 5 min at 400 g to isolate plasma, which was subsequently heat-inactivated for 30 min at 56˚C. Meanwhile, cells were washed twice with RPMI and pelleted by centrifugation for 5 min at 400 g. Complement-inactivated whole blood was reconstituted by mixing the heat-inactivated plasma with the washed cells. A control condition was also prepared, combining washed cells with untreated plasma.

For plasma killing assays, heparinized HWB was centrifuged for 10 min at 1000 g. The supernatant was recovered and filtered through a 0.2-μm membrane prior to storage at -80˚C until needed. Pools of plasma used in experiments were from ten individual donors. After thawing, plasma was systematically filtered once again through a 0.2-μm membrane. Bacteria in PBS supplemented with calcium and magnesium (Thermo Fisher Scientific, Illkirch, France) were incubated in plasma (90% final plasma concentration) at the same concentration as that used for whole blood assays ($2.25 \times 10^7$ CFU/mL). At various time points (see Figures), survival was determined by counting colonies on LB or PIA, following serial dilutions in PBS supplemented with calcium and magnesium. As for the HWB assays, the CFU count for the starting inoculum was taken as reference (100% survival) to quantify bacterial killing.

To prevent MAC formation and subsequent bacterial killing by complement, plasma was incubated 5 min at 37˚C in presence of complement-inhibitor OmCI (kindly provided by Suzan H.M. Rooijakkers and Maartje Ruyken, University Medical Center Utrecht, The Netherlands) prior to the addition of bacteria. Depending on the strain's sensitivity in plasma, OmCI was used either at 20 μg/mL (*P. aeruginosa* strains IHMA87 and PA14, *A. baumannii* and *Y. enterocolitica*) or 40 μg/mL (*P. aeruginosa* CLJ1, *B. multivorans*, EA *E. coli* 17–2 and *K. pneumoniae*). Due to aggregation of IHMA87 and to avoid artifacts of CFU counting, additional sonication step was added. Following 3h incubation in plasma with or without OmCI, 100 μL samples were transferred in 200 μL PCR tubes and placed in ultrasonic bath of the sonicator Q700 (Qsonica, Newtown, USA). Following the manufacturer instructions, tubes were placed at 2 mm from the probe, to increase the efficiency and reproducibility of the treatment. Sonication was performed at 6˚C for 45 sec at 10% intensity (5 sec pulse and 5 sec rest).

## Cyanide m-chlorophenylhydrazone treatment before plasma challenge

CCCP (Sigma-Aldrich, Saint-Quentin-Fallavier, France) solubilized in dimethyl sulfoxide (DMSO) was added to the LB culture once an $OD_{600nm}$ ~ 1 was reached. The final CCCP and

DMSO concentrations were 200 μg/mL and 0.5%, respectively. Following 1 h incubation at 37˚C with agitation (300 rpm), bacteria were recovered by centrifugation for 5 min at 5,000 g. The bacterial pellet was resuspended in PBS supplemented with calcium and magnesium. As controls, bacteria were either not treated or exposed 1h in LB with 0.5% DMSO.

## Cytokine quantification in whole blood

Cytokine concentrations in whole blood were assessed by flow cytometry. Briefly, bacteria suspended in RPMI were added to HWB ($2.25x10^7$ CFU/mL) and incubated for 3 or 6 h. Plasma was recovered following a 5-min centrifugation step at 400 g and stored at -80˚C until required. Analytes were quantified using the LEGENDplex Human Inflammation Panel 1 (Biolegend, San Diego, USA) according to the supplier's instructions, and samples were analyzed on a FACSCalibur (Becton Dickinson, Pont de Claix, France). For the non-infected control (NI), RPMI was added to the blood.

## Neutrophil and peripheral mononucleated cells isolation from blood

Circulating leukocytes were purified from heparinized blood of healthy donors as previously described [79]. Plasma was recovered following a 5 min centrifugation (room temperature, 800 g), and cells were resuspended in PBS with Trisodium citrate 1% (w/v). This suspension was carefully layered on Ficoll-Paque PLUS (GE Healthcare, Uppsala, Sweden) with a density of $1.077g/cm^3$. After a 20 min centrifugation (20˚C, 400 g), the PBMC ring was recovered and subsequently washed in RPMI 1640 medium (Thermo Fisher Scientific, Bourgoin-Jallieu, France). The cell pellet (red blood cells and granulocytes) was resuspended in cold lysis buffer (0.155M $NH_4Cl$, 10 mM $KHCO_3$ and 0.1 mM EDTA) and incubated 15 min to allow complete lysis of erythrocytes. Purified granulocytes were subsequently washed in RPMI 1640 medium. Purity (>95%) of isolated leukocytes was assessed by flow cytometry on a FACSCalibur based on FSC/SSC parameters. Viability of purified leukocytes was determined by Trypan Blue exclusion and was >95%.

## LDH release by purified leukocytes

PMNs or PBMCs ($4x10^5$ cells) were incubated with each strain of interest at a MOI of 5 for 3 h in RPMI supplemented with 10% heat-inactivated fetal calf serum on a rotating wheel at 37˚C in a 5% $CO_2$ atmosphere. Supernatants were then recovered to quantify LDH release using the cytotoxicity detection kit (Roche, Meylan, France). As a positive control (100% cytotoxicity), cells were incubated with 1%Triton X-100, while the basal level of LDH release was determined by incubating cells without bacteria.

## Time-lapse microscopy to assess the regrowth of complement-resilient bacteria

After incubation in normal plasma (or in heat-inactivated plasma for the control condition), bacteria were collected by centrifugation for 5 min at 5,000 g, washed and resuspended in PBS and then spotted on a 2% agarose pad containing LB. After complete absorption of the liquid into the pad, the preparation was sealed under a 0.17-mm glass coverslip using an adhesive plastic frame (Gene Frame 125 μL, Thermo Fisher Scientific, Illkirch, France). Time-lapse microscopy was performed at 37˚C using a 100x oil immersion objective on an inverted microscope (Axio Observer Z1, Zeiss, Germany). Phase-contrast and fluorescent images were recorded every 15 min with a Hamamatsu ORCA-Flash 4.0 digital camera and a Zeiss 38HE filterset. In each experiment performed (two for normal and two for heat-inactivated plasma),

eight isolated microcolonies were segmented using MicrobeJ (27572972) to determine their surface, growth rate and time of first division.

## Serotyping of BSI isolates

The twelve clinical isolates were serotyped using 16 monovalent antisera directed against *P. aeruginosa* LPS (Bio-Rad, Marnes-la-Coquettes, France), according to the manufacturer's instructions.

## Statistical analysis

Statistical tests were performed using SigmaPlot software. To analyze multiple comparisons, a one-way ANOVA or Kruskall-Wallis test were applied, depending on the normality of the data. A Student-Newman-Keuls pairwise comparison was then performed. Mann-Whitney U test or Student's t-test were used to compare two groups, depending of the normality of the data. Where indicated, values were log-transformed to convert initially non-normally-distributed data into a normally-distributed dataset. GraphPad Prism was used to create graphs.

## Supporting information

**S1 Fig. All tested strains except YIK have a similar capacity to induce cytokine release.** Strains, as indicated, were incubated in HWB (n = 6, from different donors) for the time points indicated, and cytokines were quantified by a multiplex analysis. (*): different from T3h+YIK. (&): different from T6h+YIK. (#): different from the cognate T3h. Mann-Whitney test: # $p < 0.05$. Kruskal-Wallis test, $p < 0.05$; Student-Newman-Keuls post-hoc test: * and &, $p < 0.05$. (TIF)

**S2 Fig. All strains, except YIK, display similar levels of cytotoxicity toward circulating leukocytes.** Cytotoxicity of indicated strains on purified neutrophils or mononucleated cells was analyzed by LDH release (n = 3, from different donors). Following a 3-h incubation of leukocytes with bacteria at a MOI of 5, LDH released by dead cells was quantified from the supernatants. The LDH release obtained following treatment with 1%Triton X-100 correspond to 100% cytotoxicity. (&): different from PA7. (#): different from YIK. Kruskal-Wallis test, $p < 0.05$; Student-Newman-Keuls post-hoc test: & and #, $p < 0.05$. Note that none of the strains tested induced a significantly different killing of mononucleated cells. (TIF)

**S3 Fig. The presence of evaders is not due to a limited complement activity. (A)** Following a first challenge, the plasma is still bactericidal. Schematic view of the experimental procedure: PA14 ($22.5 \times 10^6$ CFU/mL) was incubated in a pool of plasma for 3 h. Following the first challenge, the plasma was sterilized by passing through a 0.2-μm filter, and re-challenged with the same number of PA14 cells for 1 h at 37°C. More than $20 \times 10^6$ bacteria/mL were eliminated during this second incubation, showing that the plasma is still bactericidal after 3 h of contact with the bacteria. **(B)** Reducing the bacterial load does not affect the proportion of surviving cells. PA14 was incubated in a pool of plasma at concentrations of $22.5 \times 10^6$ (white) or $2.25 \times 10^6$ (grey) CFU/mL, and survival was assessed following incubation for 3 h. The proportions of evaders were unchanged regardless of the initial bacterial load. Data represent mean ± SD of three independent experiments (**A** and **B**). (TIF)

**S4 Fig. Plasma evaders are either solely or partly composed of complement-tolerant cells.** Depending on the species, additional fluid-phase bactericidal effectors other than complement

are likely involved in the selection of evaders. Bacteria were incubated in a pool of human plasma for 3 h with (hashed) or without (solid) OmCI at 20 μg/mL (*A. baumannii* and *Y. enterocolitica*) or 40 μg/mL (*B. multivorans*, *E. coli* 17–2 and *Y. enterocolitica*) and survival was measured based on CFU counts. Data represent mean ± SD of three independent experiments. (TIF)

**S5 Fig. Growth curves of bacteria used in this study.** Growth kinetics in LB of the six *P. aeruginosa* strains CLJ1, IHMA, PA7, PAO1, PA14, YIK, *E. coli* CF7968 **(A)**, and the other Gram-negative species used in this study **(B)**. Data represent mean ± SD of two independent cultures **(A** and **B)**. (TIF)

**S6 Fig. CCCP treatment efficiently shuts down bacterial metabolism but is non-toxic for the bacterial cells.** Exponentially growing IHMA87 cultures in LB were either not treated (solid), treated 1 h with CCCP at 200 μg/mL (dotted) or exposed to 0.5% DMSO for 1 h (hashed). Cells were subsequently recovered and incubated in LB alone. Evolution of the CFU number was monitored by plating after 1 h, 2 h and 3 h. Bacterial survival is unaffected following CCCP treatment, however treated cells are still non-growing even 3 h after removing the stress. At 0.5%, DMSO does not affect bacterial growth. Data represent mean ± SD of three independent experiments. (TIF)

**S1 Table. Bacterial strains used in this work.** (PDF)

**S2 Table. Patient demographics from which clinical BSI strains were isolated.** (TIF)

## Acknowledgments

We are grateful to Suzan H.M. Rooijakkers and Maartje Ruyken (University Medical Center Utrecht, The Netherlands), for providing OmCI. We acknowledge Thomas Hindré (University Grenoble Alpes, France), Pierre Marcoux (LETI, CEA Grenoble, France) and Eric Cascales (CNRS, Aix-Marseille Université, France) for gift of *E. coli* CF7968, *K. pneumoniae* NCTC96 and enteroaggregative *E. coli* 17–2, respectively. *P. aeruginosa* strain IHMA879472 was kindly provided by International Health Management Association (IHMA; USA). We are grateful to Peter Panchev for English editing of the initial manuscript, and to Maighread Gallagher-Gambarelli (TWS Editing) for suggestions on language usage for the final manuscript. We thank undergraduate student Sean Moro (University Grenoble Alpes, France) for his help in assessing the cytotoxicity of the different strains on purified blood leukocytes.

## Author Contributions

**Conceptualization:** Stéphane Pont, Nathan Fraikin, Laurence Van Melderen, Ina Attrée, François Cretin.

**Formal analysis:** Stéphane Pont, Ina Attrée, François Cretin.

**Funding acquisition:** Laurence Van Melderen, Ina Attrée.

**Investigation:** Stéphane Pont, Nathan Fraikin, Yvan Caspar, François Cretin.

**Project administration:** Ina Attrée, François Cretin.

**Resources:** Yvan Caspar.

**Supervision:** Laurence Van Melderen, Ina Attrée, François Cretin.

**Writing – original draft:** Stéphane Pont, Ina Attrée.

**Writing – review & editing:** Stéphane Pont, Nathan Fraikin, Yvan Caspar, Laurence Van Melderen, Ina Attrée, François Cretin.

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
