## [Decision Letter · Decision Letter 0]

18 Sep 2020

Dear Dr. Attree,

Thank you very much for submitting your manuscript "Bacterial behavior in human blood reveals complement evaders with persister-like features" for consideration at PLOS Pathogens. As with all papers reviewed by the journal, your manuscript was reviewed by members of the editorial board and by several independent reviewers. In light of the reviews (below this email), we would like to invite the resubmission of a significantly-revised version that takes into account the reviewers' comments.

The reviewers raised a significant issue of whether the effects observed can be ascribed to evasion of complement rather than another heat labile mechanism of bacterial killing in plasma? They suggest some approaches to inhibit complement killing, and experiments could be performed with Mg2+/EGTA.

We cannot make any decision about publication until we have seen the revised manuscript and your response to the reviewers' comments. Your revised manuscript is also likely to be sent to reviewers for further evaluation.

Sincerely,

Christoph Tang

Section Editor

PLOS Pathogens

Christoph Tang

Section Editor

PLOS Pathogens

Kasturi Haldar

Editor-in-Chief

PLOS Pathogens

orcid.org/0000-0001-5065-158X

Michael Malim

Editor-in-Chief

PLOS Pathogens

orcid.org/0000-0002-7699-2064

Reviewer's Responses to Questions

**Part I - Summary**

Reviewer #1: The paper describes a very interesting finding - the discovery of “evaders”, phenotypic variants of bacterial cells that survive in blood. Most pathogen cells in blood are killed by a combination of complement and phagocytosis, but the authors of this study find that killing of P. aeruginosa and other pathogens in blood in biphasic, with a subpopulation of surviving evaders. They provide a good analysis of the obvious factors that may contribute to survival, such as the presence of toxins in the pathogens, but find that these are not involved in survival. Importantly, the authors show that this phenotypic subpopulation is different from persisters, since there are more of such cells in a growing population as compared to stationary (more persisters in stationary). Survival was also energy-dependent, suggesting the presence of an active process. It appears that variation in expression of some complement-protection factors confers resistance. This study sets the stage for identifying the mechanism of this interesting and important phenomenon. The study is well-exceuted, experiments are simple and straightforward, with considerable statistical significance. Minor comments follow.

The abstract lacks any mechanistic information and does not do justice to this work. Please add information in the abstract related to specific factors examined in this study for their ability to contribute to survival, such as toxins and TIII secretion system.

A very interesting isolate that was completely resistant to killing was not pursued. This is understandable, given the focus of the study on evaders, but it would be useful to note if there are known precedents of such isolates, and whether the mechanism for full resistance to blood has been described.

There is a positive correlation between the level of survival of the bulk population and the level of evaders this population produces, makes sense to point this out.

Reviewer #2: The manuscript of Pont et al., studies how P. aeruginosa can persist within the blood. With a panel of different P. aeruginosa strains it becomes clear that a small population survives in whole blood. This effect is also observed in plasma and can be heat inactivated. Since, complement is able to kill Gram-negative bacteria by forming pores in the membrane the surviving population is called ‘complement evaders’. It is an interesting and relevant finding that a small part of the bacteria manages to survive in plasma. The complement evaders show sensitivity to plasma again after reexposure to plasma indicating that these are phenotypic variants. The authors evaluate if the complement evaders have similar characteristics as persisters. In contrast to persisters, the complement evaders are not more abundant in stationary phase cultures and experiments suggested that metabolic activity is required.

The findings in this manuscript are important in the understanding how bacteria persist within the blood. The results will lead to follow-up studies for more detailed understanding how the described subpopulation survives in blood. The article would benefit from extra experiments that more definitely show that the bacteria indeed are persistent to complement. Plasma contains a lot of proteins that can have an effect on the survival of bacteria. Heat inactivation is not sufficient since this inactivates a lot more proteins than components of the complement system.

Reviewer #3: This study seeks to better understand the ability of the bacteria, Pseudomonas aeruginosa, to survive and cause serious infection in the human blood stream. To do this, they assay the ability of six strains (3 lab strains, & 3 clinically isolated strains) to survive in human whole blood (HWB), revealing significant variability between and within strains in the ability to survive complementation. These differences were not accounted for by genetic differences and so the study has the potential to make an important contribution to our understanding of our particular vulnerability to this kind of infection, and phenotypic defence mechanisms of bacteria more generally.

A subpopulation of survivors was apparent in all of the tested P. aeruginosa strains leading the authors to speculate that these HWB ‘evaders’ may be similar to bacterial persisters (defined in the literature as phenotypic variants within a bacterial population that are able to transiently survive high antibiotic doses, while remaining genetically identical to other cells within the population). To explore the potential role of persistence in HWB-treated survival, the authors conducted a series of killing curves to test for the expected biphasic killing curves, as seen with persister cells within the literature. The killing curves showed that all strains showed similar phenotypic variation in response to HWB.

After detecting complement evaders in all P. aeruginosa strains, the authors expanded their assay to include 7 further species of Gram-negative bacteria. Two species demonstrated non-biphasic killing curves in HWB, however, the remaining five showed distinct killing curves.

To further explore the potential for ‘evaders’ to be persister cells, the authors carried out assays on log versus stationary growing cells, since the literature has highlighted that persister cells predominantly emerge during stationary phase. Contrary to expectation given the literature, three P. aeruginosa strains and two of the selected Gram-negative species generated a greater proportion of ‘evaders’ in exponential phase. Studies already conducted on persister formation have highlighted the roll of cell dormancy in antibiotic survival. Therefore, the authors explored the role of inhibiting metabolic activity with varying results. Lastly, the different strains and species were observed by microscopy to evaluate division rates after incubation with plasma and heat-inactivated plasma. No significant difference was observed between the heat-treated/active plasma growth rates. Taken together, the authors deduced that metabolism played a key role in ‘evader’ formation.

Strength: this is a detailed and fascinating study of a clinically important yet under-studied bacterial behaviour. The paper is clearly written and easy to follow (thank you!)

Weakness: evidence for evasion being a distinct behaviour from persistence was not so convincing and not necessary for the paper to make an important contribution.

**Part II – Major Issues: Key Experiments Required for Acceptance**

Reviewer #1: none

Reviewer #2: Required:

They should show that the bacteria indeed persist complement and not other heat-sensitive bactericidal components in serum

Other major issues:

The term persister-like that is used in the title is not fitting that well, since according to the conclusion complement evaders behave differently.

In figure 2, the percentage of surviving cells is higher (10-100 fold) for the bacteria CJL1, IHMA and PA14 in whole blood compared to plasma. How do the authors explain this difference? Is the number of complement evaders different in whole blood? Is this an effect of the cells that are present?

In figure 2B, additional controls are required to claim that the observed effect is dependent on killing by the membrane attack complex. Since plasma contains a lot of proteins that can have an effect on the survival of bacteria it is necessary to add specific MAC inhibitors (OmCI, Eculizumab). Heat inactivation is not sufficient since this inactivates a lot more proteins than components of the complement system.

Figure S2 is in its current form not informative. The authors should include a cell death stain marker to determine if there is a cytotoxic effect. In addition, percentage of CD45 cells should be changed. Quantification of the percentage of dead (purified) neutrophils or PBMCs would be more informative. In the current graphs, the counts of both populations can change and thereby affecting the percentage which makes it hard to interpret the data. Furthermore, in the treated samples events with a lower forward and side scatter are partly outside the gate, that is used for analysis. The cytotoxic data are not required for the manuscript, since the focus is complement evaders.

In Figure 3B, the timepoints differ, for some strains there is a timepoint at 1 hour and for some after 2 hours. With the current graph it is hard to judge if there is a biphasic killing curve. Therefore, a few more timepoints and one after 6 hours would be informative to judge if reexposures of the complement evaders result in the same graph.

The authors should check that complement activation is similar on stationary and exponential phase bacteria. The expression of LPS O-antigen, capsule and outer membrane proteins changes which are important in complement activation. This is important in the comparison of the number of evaders between exponential and stationary phase. An alternative explanation could be that complement is less well activated on stationary vs exponential phase bacteria.

The complement evaders survive in plasma for a prolonged time. This study would benefit by analyzing complement activation on the evader cells. For instance, check the C3b, C5b-9 levels of the surviving cells by flow cytometry (after 3-6 hours). Is the complement activation and membrane damage the same, but can the evaders deal with the membrane damage in contrast to normal cells?

In figure 4B the effect of CCCP indicates that metabolic activity is required for complement evaders. CCCP itself does not kill untreated cells. The data indeed point to the direction that a membrane potential is required. Were the control cells also treated with 0.5 % DMSO, because this was not clear in the legend and methods. This would be important since DMSO can also affect bacteria. In addition, the statement that metabolic activity is required for complement evaders is only based on the CCCP result. Alternative approaches would help to convince that metabolic activity is required and that this is different compared to persisters (e.g. treatment with bacteriostatic antibiotics).

Reviewer #3: We (I am writing this review with my PhD student, so we are a reviewer team) were concerned that the evidence for evaders to be difference from persister cells was interpreted too strongly, especially in the light of continuing controversy over what persisters really are and how to categorise them. (The importance of the paper’s potential contribution does not rest on there being a distinction in our view.)

We, therefore suggest the claim for them to be different is qualified in light of recent literature: there are perhaps too many similarities between persisters and ‘evaders’ for ‘evaders’ to be defined as a distinct physiological state. Balaban et al., 2019 explore the notion of spontaneous versus triggered persisters. There is some indication that spontaneous persisters arise during exponential growth, therefore, we may expect metabolism to affect persister proportion during exponential growth. The distinct complement evasion strategy is an interesting notion within this study, though it should be explored further before any distinction between persisters and ‘evaders’ can be made. There is perhaps scope to assay the strains with a common antibiotic to compare with HWB.

**Part III – Minor Issues: Editorial and Data Presentation Modifications**

Reviewer #1: The abstract lacks any mechanistic information and does not do justice to this work. Please add information in the abstract related to specific factors examined in this study for their ability to contribute to survival, such as toxins and TIII secretion system.

A very interesting isolate that was completely resistant to killing was not pursued. This is understandable, given the focus of the study on evaders, but it would be useful to note if there are known precedents of such isolates, and whether the mechanism for full resistance to blood has been described.

There is a positive correlation between the level of survival of the bulk population and the level of evaders this population produces, makes sense to point this out.

Reviewer #2: Line 54 and 340 “while genetically identical” and “do not harbor genetic mutations”

With this claim you expect that the surviving bacteria (complement evaders) were sequenced and are identical to the original strain. Please add the data or state that this is assumed because reexposure to plasma of the evaders result in a similar percentage of survivors as observed after the initial plasma treatment.

Line 167, hypersensitive change to sensitive

Figures S1 does not add much information to the manuscript.

Add a detection limit in the graphs of % survival, it is now unclear when there are no surviving cells at all. Sometimes these points are just removed like for PaG8 in figure 5B. Please add this to all graphs.

Figure 2C significance is not correctly indicated, should be significant between control and cytochalasin D

Legend of figure 3, description jumps from A to C.

Line 321-323 “to fully resist in whole blood strains had to display cytotoxicity and complement resistance. This statement is only based on a single strain (Yfk), which is not analyzed in detail. More strains and detailed analysis should be performed for this claim.

It is for some experiments not clear how the bacteria were cultured (only stated exponential phase). Overnight bacteria were diluted and shaken until OD600. Please add some more details, diluted x times in.. at 37C. How is the number of bacteria quantified (since 2.25 E7 bacs/ml are used)?

Figures S3A, it is not clear how this is performed and would be important to compare the bactericidal activity of plasma after 3 hours incubation to fresh plasma. The other option would be to assay the bactericidal activity with a classical complement assay (e.g. CH50)

Figure S3B. It would be informative to test the influence of the bacterial concentration over a wider range.

Line 308-310, statement about exotoxins. The authors did not check if the toxins were expressed and did not test a panel of strains with these toxins. Based on the data shown it is hard to determine that the presence of toxins is unrelated to the strains capacity to survive in blood.

Line 345-346 describes the complement resistant K. pneumoniae, would be informative if this comes back in the discussion. Was this selection of a subpopulation (e.g. heteroresistant) or probably a mutation that causes this different phenotype?

Legend fig S5 – Check description, it states plating for 2 hours, should be incubation.

Reviewer #3: Lines 479-481: On average, how many cells were observed? Were there only two replicates for each strain and/or species? This will affect how reliable the observations were as at least three replicates should have been used

Line 491: ‘depending of the normality’ should be ‘depending on…’

Figure 2: The patterns and data points and colours used on the graph are a bit nasty to look at. The added data point circles impair boxplot visualisation.

Figure 3: The order of the lettering in the legend is unclear. ‘A)’ goes to ‘C)’, which then goes to ‘B) and D)’. Could these figures be re-ordered, or the legend order altered?

PLOS authors have the option to publish the peer review history of their article (what does this mean?). If published, this will include your full peer review and any attached files.

Reviewer #1: **Yes: **Kim Lewis

Reviewer #2: No

Reviewer #3: No
---

## [Editor Report · Decision Letter 1]

3 Nov 2020

Dear Dr. Attree,

We are pleased to inform you that your manuscript 'Bacterial behavior in human blood reveals complement evaders with some persister-like features' has been provisionally accepted for publication in PLOS Pathogens.

Best regards,

Christoph Tang

Section Editor

PLOS Pathogens

Kasturi Haldar

Editor-in-Chief

PLOS Pathogens

orcid.org/0000-0001-5065-158X

Michael Malim

Editor-in-Chief

PLOS Pathogens

orcid.org/0000-0002-7699-2064
---

## [Editor Report · Acceptance letter]

1 Dec 2020

Dear Dr. Attree,

We are delighted to inform you that your manuscript, "Bacterial behavior in human blood reveals complement evaders with some persister-like features," has been formally accepted for publication in PLOS Pathogens.

Best regards,

Kasturi Haldar

Editor-in-Chief

PLOS Pathogens

orcid.org/0000-0001-5065-158X

Michael Malim

Editor-in-Chief

PLOS Pathogens

orcid.org/0000-0002-7699-2064